# Osa-miR11117 Targets *OsPAO4* to Regulate Rice Immunity against the Blast Fungus *Magnaporthe oryzae*

**DOI:** 10.3390/ijms242216052

**Published:** 2023-11-07

**Authors:** Shang Gao, Yan Hou, Qiwei Huang, Pengzhi Wu, Zhikai Han, Danhong Wei, Huabin Xie, Fengwei Gu, Chun Chen, Jiafeng Wang

**Affiliations:** National Plant Space Breeding Engineering Technology Research Center, Guangdong Key Laboratory of Plant Molecular Breeding, South China Agricultural University, Guangzhou 510642, China; 15038190778@163.com (S.G.); h1149130339@163.com (Y.H.); huangqiweihqw@stu.scau.edu.cn (Q.H.); pengzhiw220@gmail.com (P.W.); zhikai@stu.scau.edu.cn (Z.H.); 17718844744@163.com (D.W.); 20221015020@stu.scau.edu.cn (H.X.); 99g999999ufw@gmail.com (F.G.); chchun@scau.edu.cn (C.C.)

**Keywords:** Osa-miR11117, *OsPAO4*, *Magnaporthe oryzae*, microRNA, polyamine

## Abstract

The intricate regulatory process governing rice immunity against the blast fungus *Magnaporthe oryzae* remains a central focus in plant–pathogen interactions. In this study, we investigated the important role of Osa-miR11117, an intergenic microRNA, in regulating rice defense mechanisms. Stem-loop qRT-PCR analysis showed that Osa-miR11117 is responsive to *M. oryzae* infection, and overexpression of Osa-miR11117 compromises blast resistance. Green fluorescent protein (GFP)-based reporter assay indicated *OsPAO4* is one direct target of Osa-miR11117. Furthermore, qRT-PCR analysis showed that *OsPAO4* reacts to *M. oryzae* infection and polyamine (PA) treatment. In addition, *OsPAO4* regulates rice resistance to *M. oryzae* through the regulation of PA accumulation and the expression of the ethylene (ETH) signaling genes. Taken together, these results suggest that Osa-miR11117 is targeting *OsPAO4* to regulate blast resistance by adjusting PA metabolism and ETH signaling pathways.

## 1. Introduction

Pathogens such as bacteria, fungi, and oomycetes frequently attack plants, prompting the plants to evolve a sophisticated defense system comprising pathogen-associated molecular pattern (PAMP)-triggered immunity (PTI) and effector-triggered immunity (ETI) to fight against the invaders [1]. Accumulating evidence has revealed that plant microRNAs (miRNAs) are critical regulators in both PTI and ETI responses against pathogens [2,3]. miRNAs are a type of endogenous RNA (21~24 nucleotides) that consist of a single strand and play a crucial role in regulating gene expression at the post-transcriptional level or the translation level [4]. Importantly, miRNAs play a pivotal role as significant regulators of diverse cellular and biological processes, such as plant growth, development, reproduction, and responses to both biotic and abiotic stresses [5,6]. Under the invasion of pathogens, miRNA expression in some plants changed significantly. Differentially expressed miRNAs may play an important role in plant disease resistance by inhibiting or activating the expression of some target genes, regulating plant signal transduction pathways, and activating key factors of the disease resistance pathway [7,8]. The precise function of a given miRNA depends on the specific target genes and their involvement in various biological processes. For example, miR393, the first discovered miRNA in *Arabidopsis*, regulates the host immunity against *Pseudomonas syringae* DC3000 through inhibiting auxin signaling by targeting the auxin receptors, including *TIR1*, *AFB2*, and *AFB3* [9]. Furthermore, it has been identified that miR160a, miR398b, and miR773 play a crucial role in the modulation of callose deposition in PTI signaling [10]. In addition, the miR482/miR2118/miR472 superfamily possesses the ability to regulate the expression of NBS-LRR genes to prevent the excessive expression of these genes when there is no pathogen present, indicating the critical roles of miRNA in ETI signaling [11,12].

Polyamines (PAs) are a group of amine molecules, mainly including putrescine (Put), spermidine (Spd), spermine (Spm), and thermospermine (T-Spm), that exist widely across various living organisms. There is evidence pointing to the significant role of PAs in the interactions between plants and pathogens [13]. The levels of PAs and the activity of enzymes involved in PA metabolism were observed to increase upon infection of plant tissues by both biotrophic and necrotrophic pathogens [14,15]. Reduced Put levels in *adc* loss-of-function mutants and *adc*-silenced lines lead to increased susceptibility to pathogens [16,17]. Conversely, the overexpression of the *AtADC2* gene or supplementation of Put in *Arabidopsis* induces the expression of plant defense genes and promotes local salicylic acid biosynthesis [16]. In addition to Put, Spm is also regarded as a significant signaling molecule that regulates resistance to bacterial pathogens and viruses [11,18]. Similarly, reduced T-Spm (a structural isomer of Spm) levels in cotton *acl5*-silenced plants lead to susceptibility to *Verticillium dahliae* infection [19]. Moreover, the pathogenic bacterium *Ralstonia solanacearum* produces Brg11 effector protein to target the tomato *ADC1* and *ADC2* genes and promote Put accumulation [20]. In response to plant pathogens such as *P. syringae*, there is an increase in polyamine oxidase (PAO) activity, leading to Spm secretion and H_2_O_2_ accumulation in the apoplast, which contributes to the hypersensitive response (HR) [21]. In tobacco, inhibiting the expression of the gene coding for apoplastic PAO prevented PA degradation, cryptogein-induced apoplastic H_2_O_2_ generation, and cell death [22]. In addition to its cytotoxic effect, PAO-generated H_2_O_2_ also has a signaling role in the HR [22]. Although the enzyme characteristics of all *Arabidopsis* PAOs and most rice PAOs have been determined, their biological functions in plant defense mechanisms against pathogens still remain ambiguous.

Rice is a crucial food source for more than half of the global population, but its production is often threatened by a variety of diseases, including rice blast caused by *M. oryzae*. In recent years, miRNAs have been identified as key regulators during rice responsive to *M. oryzae* infection. For instance, miR398b was found to enhance rice immunity against *M. oryzae* by targeting multiple SOD genes to boost H_2_O_2_ accumulation [23]. In contrast, miR164a [24], Osa-miR167d [25], miR168 [26], miR169 [27], miR319b [28], miR396 [29], miR444b.2 [30], miR530 [31], Osa-miR535 [32], miR1432 [33], Osa-miR1871 [34], and Osa-miR1873 [35] exert a negative regulatory influence on rice’s immune response to *M. oryzae*. Conversely, Osa-miR159a [36], miR160a [37], Osa-miR162a [38], miR166k-miR166h [39], and miR7695 [40] have been identified as contributors to enhancing rice immunity against *M. oryzae.*

In our previous study, we identified a number of miRNAs that were differentially responsive to *M. oryzae* [41]. Among them, Osa-miR11117 (previously named miR-T21) was differentially accumulated in the susceptible and resistant rice accessions upon *M. oryzae* infection, indicating Osa-miR11117 might be involved in the regulation of rice immunity against *M. oryzae*. In this study, we explored Osa-miR11117 and its target *OsPAO4* in rice defense against *M. oryzae*. First, Osa-miR11117 responded to *M. oryzae* and displayed varying patterns in susceptible and resistant rice lines. Overexpression of Osa-miR11117 increased susceptibility, while reduced expression led to some resistance. Second, *OsPAO4*, identified as a target gene of Osa-miR11117, played a crucial role in rice’s defense. Third, *OsPAO4* responded to Spm and Spd, hinting at its involvement in PA catabolism. Last, knocking out *OsPAO4* made rice more susceptible, while overexpression of *OsPAO4* enhanced resistance and impacted polyamine levels and the ethylene signaling pathway. This study highlights that Osa-miR11117 targets *OsPAO4s* to modulate rice blast resistance via polyamine metabolism and ethylene pathways.

## 2. Results

### 2.1. Osa-miR11117 Is Responsive to Blast Fungus

In earlier investigations, we identified several novel miRNA candidates that were responsive to *M. oryzae* in rice [41]. Herein, we focused on one such miRNA candidate, miR-T21, in Dong et al. [41] and conducted further investigations. Upon searching in miRBase, it was observed that the mature sequence of miR-T21 is relatively conserved, showing significant homology with miRNAs from various species, including lja-miR11117a-3p/lja-miR11117b-3p from lotus, hsv2-miR-H6-3p from herpesvirus, gra-miR7494b from canary, and hsa-miR-5196-3p from humans (Appendix A). Due to the high similarity of miR-T21 with lja-miR11117a-3p/lja-miR11117b-3p, herein, we rename ‘miR-T21’ as ‘Osa-miR11117’. Osa-miR11117 is a 23-base intergenic miRNA located on the positive strand at the end of the long arm of rice chromosome 2, between the genes *LOC_Os02g35840* and *LOC_Os02g35860* (Figure 1A). UNAFold analysis [42] predicted that it could be folded into a typical miRNA precursor stem-loop structure (Figure 1B). The expression of Osa-miR11117 was confirmed in resistant accession (Pik H4 NIL) by Northern blot (Figure 1C). Analysis of the Osa-miR11117 *GUS* promoter activity indicates that it is mainly expressed in the coleoptile and leaves, with low expression in the young roots, but not in the stems and internodes (Figure 1D), indicating the expression of Osa-miR11117 is tissue-specific. Furthermore, using stem-loop qRT-PCR analysis, it was found that Osa-miR11117 displays a different expression pattern in the susceptible (LTH) and resistant accessions in response to blast fungus infection (Figure 1E). Upon *M. oryzae* invasion, the amounts of Osa-miR11117 were significantly increased at 6 h and 72 h post-inoculation (hpi) in LTH but kept unchanged at 12–48 hpi (Figure 1E). In contrast, the accumulation of Osa-miR11117 was increased dramatically at 6 and 24 hpi in Pik H4 NIL but decreased at 12, 48, and 72 hpi (Figure 1E). These results indicated that Osa-miR11117 is responsive to *M. oryzae* infection and functions in rice immunity to blast fungus.

### 2.2. Overexpression of Osa-miR11117 Enhances Susceptibility to Infection by M. oryzae

To investigate the impact of Osa-miR11117 on rice blast resistance, we generated transgenic lines with increased expression of the miRNA (OX11117) and the target mimic of the miRNA (MIM11117), and CRISPR/Cas9-mediated mutagenesis of Osa-miR11117 (KO11117) under the Pik-H4 NIL background, respectively. Three lines of each transgenic line (MIM11117 and KO11117) that showed decreased expression of Osa-miR11117 and OX11117 that showed increased expression of Osa-miR11117 were selected for subsequent analysis (Figure 2A). Next, we examined whether the overexpression or knockout of Osa-miR11117 has a consequence in blast disease resistance. Two-week-old leaves of OX11117, MIM11117, KO11117, and wild-type plants were punch-inoculated with *M. oryzae* GDYJ7 spores, and disease symptoms were followed over time. Compared with wild-type plants, OX11117 plants consistently exhibited higher susceptibility to infection by *M. oryzae*, as evidenced by disease lesions and relative fungal biomass compared to the control, while MIM11117 and KO11117 lines showed a little resistance to *M. oryzae* (Figure 2B–D). These findings suggest that overexpression of Osa-miR11117 can compromise blast resistance.

### 2.3. OsPAO4 Is One of the Target Genes of Osa-miR11117

To identify the target genes of Osa-miR11117, we used the online tool psRNATarget website (https://bio.tools/psrnatarget accessed on 12 March 2019) and predicted the eight most potential targets with strict parameters (Appendix A). To further examine which predicted genes are the target genes of Osa-miR11117, a green fluorescent protein (GFP)-based reporter assay in *Nicotiana benthamiana* plants was performed. Briefly, two constructs, including 35S: target gene-GFP and 35S: Osa-miR11117, were coinjected into *N. benthamiana* leaves for further observation. After examining these candidate genes with the GFP-based reporter assay, the protein expression of two candidate genes (*LOC_Os04g57550* and *LOC_Os05g51380*) was inhibited by Osa-miR11117 (Appendix A). *LOC_Os04g57550*, encoding the polyamine oxidase 4 (*OsPAO4*) in rice, was selected for further study. To further verify that Osa-miR11117 indeed downregulates the expression of *OsPAO4*, we conducted the GFP-based reporter assay with two constructs, including *35S:OsPAO4ts-GFP* and *35S:OsPAO4mts-GFP*. The former two contained the putative target site of *OsPAO4* and a mutated target site, respectively (Figure 3A,B). Coexpression of Osa-miR11117 and *35S:OsPAO4ts-GFP* resulted in a noticeable reduction in GFP intensity and protein concentration in *N. benthamiana* leaves compared to expression of the construct alone (Figure 3C,D). In contrast, coexpression of *35S:OsPAO4ts-GFP* alone or *35S:OsPAO4mts-GFP* with Osa-miR11117 did not lead to any changes in *GFP* expression or protein concentration (Figure 3C,D). The expression level of *OsPAO4* in the transgenic lines (OX11117, MIM11117, and KO11117) was further validated. Compared to the wild type, *OsPAO4* is significantly enhanced in both KO11117 and MIM11117 transgenic plants (Figure 3E). Interestingly, in OX11117 transgenic plants, the expression of *OsPAO4* is also highly increased (Figure 3E), strongly suggesting that *OsPAO4* is very likely a target gene of Osa-miR11117, and Osa-miR11117 might inhibit the expression of *OsPAO4* at translation level.

### 2.4. OsPAO4 Is One M. oryzae and PA Responsive PAO Gene

To further explore the involvement of *OsPAO4* in rice blast resistance, we examined the amounts of *OsPAO4* in the susceptible accession (LTH) and the resistant accession (Pik-H4 NIL). Pik-H4 NIL carries a resistance (*R*) gene *Pik-H4* and exhibits resistance to *M. oryzae* isolates (GDYJ7) carrying *Avr-PikE*. After inoculating GDYJ7 on LTH and Pik-H4 NIL leaves, the abundance of *OsPAO4* decreased first (6 and 12 hpi) and increased after 24 hpi in LTH (Figure 4A). Pik-H4 NIL also decreased at 6 and 12 hpi but increased more at 48 and 72 hpi compared to LTH, which is a little contrary to the expression level of Osa-miR11117 in response to *M. oryzae* infection (Figure 1E), suggesting the involvement of *OsPAO4,* a target of Osa-miR11117, in the rice blast resistance process.

There are seven *PAO* genes in rice [43]. Among them, *OsPAO4* is closed to *OsPAO5*, and they are both closed to *AtPAO4* (Appendix A). Converse to *OsPAO2, OsPAO6,* and *OsPAO7*, the other four *PAO* genes (*OsPAO1*, *OsPAO3*, *OsPAO4,* and *OsPAO5*) expressed in a higher level in rice leaves (Appendix A). Interestingly, only *OsPAO4* is responsive to *M. oryzae* (Figure 4A and Appendix A). Further, semi-quantitative RT-PCR and qRT-PCR assay indicated that *OsPAO4* expresses a high level in callus and root, indicating the important roles of *OsPAO4* in rice growth and development (Appendix A).

It has shown that the products of *OsPAO3*, *OsPAO4,* and *OsPAO5* localize to peroxisomes, and they catalyze the polyamine back-conversion reaction [44]. The role of *OsPAO4* expression in response to PAs was further examined. As shown in Figure 4B, PAs treatment did not significantly affect the expression of *OsPAO4* in the susceptible accession. Interestingly, after PAs treatment, both Spm and Spd significantly increased the expression of *OsPAO4,* while Put did not, in resistant accession (Figure 4C). These results indicated that *OsPAO4* may be involved in rice blast resistance through the Spm and/or Spd catabolism pathway in blast-resistant accession.

### 2.5. Knockout of OsPAO4 Enhances the Susceptibility to Infection by M. oryzae

To investigate the involvement of *OsPAO4* in rice blast resistance, *ko-ospao4* transgenic lines were generated under Pik-H4 NIL background using CRISPR/Cas9 gene editing technology. Three lines with different mutation sites were selected for further study. The *ko-ospao4* lines enhanced the susceptibility to infection by *M. oryzae* with larger lesions (Figure 5A,B) and more fungal biomass than the WT control by punch methods (Figure 5C). Further, qRT-PCR assay showed that the expression of *PR1a*, *PR1b,* and *PR10a* was significantly reduced in the *ko-ospao4* transgenic lines (Figure 5D), which is consistent with their decreased disease resistance. We then measured H_2_O_2_ contents and the activities of the H_2_O_2_-scavenging enzymes, such as superoxide dismutase (SOD), peroxidase (POD), and catalase (CAT) in WT and *ko-ospao4* lines. The *ko-ospao4* lines had significantly lower H_2_O_2_ contents than the wild type (Figure 5E), indicating that less H_2_O_2_ accumulated in *ko-ospao4* lines. In addition, SOD, POD, and CAT activities were significantly higher in the *ko-ospao4* lines than in the wild type (Figure 5F–H), indicating that the active oxygen scavenging ability was enhanced in *ko-ospao4* lines. These results indicate that less ROS accumulated in *ko-ospao4* leaves than in the wild type. H_2_O_2_ contents of *ko-ospao4* and WT leaves were further measured after inoculation by *M. oryzae*. Compared to the wild type, H_2_O_2_ contents were significantly reduced in *ko-ospao4* leaves (Figure 5I). These above findings suggest that the knockout of *OsPAO4* enhances the enzyme activity in general and weakens the disease resistance of rice to blast fungus.

### 2.6. Overexpression of OsPAO4 Enhances the Resistance against M. oryzae

To further investigate whether *OsPAO4* positively contributes to rice blast resistance, *OsPAO4* over-expressing transgenic plants were generated under Pik-H4 NIL background, detached leaves from *OsPAO4* over-expressing plants were challenged by punching inoculation with Guy11 (a compatible strain to Pik-H4 NIL line) spores. The leaves of *OsPAO4* over-expressing plants developed mild disease symptoms (Figure 6A), manifested by significantly smaller lesion sizes (Figure 6B) and less fungal biomass on transgenic leaves compared to that from WT plants (Figure 6C), which is inconsistent with further inoculation assay using spray methods (Figure 6D). As opposite to *ko-ospao4* transgenic lines (Figure 5D), the expression of *PR1a*, *PR1b,* and *PR10a* was significantly increased in *OsPAO4* over-expressing plants (Figure 6E), which is consistent with the enhanced disease resistance (Figure 6A–C). As expected, the *OE-OsPAO4* lines exhibited notably elevated H_2_O_2_ content in comparison to the wild type (Figure 6F). Furthermore, the activities of SOD, POD, and CAT were substantially diminished in the *OE-OsPAO4* lines as opposed to the wild type (Figure 6G–I). Hence, these collective observations strongly imply that overexpression of *OsPAO4* can improve rice blast resistance by regulating the level of H_2_O_2_.

### 2.7. OsPAO4 Regulates Rice Blast Resistance by Controlling Both the Accumulation of PA and the Ethylene Signaling Pathway

Due to its potential role in regulating the levels of H_2_O_2_ and its contribution to rice blast disease resistance, the function of *OsPAO4* in PA accumulation was investigated in blast-resistant accession (Pik-H4 NIL). As shown in Figure 7A–C, knockout of *OsPAO4* or overexpression of *OsPAO4* can both lead to accumulation of Spm and Spd, but not the Put. Furthermore, knockout of *OsPAO4* resulted in a significantly higher accumulation of Spd and Spm compared to the wild type, suggesting that changes in *OsPAO4* expression levels affected polyamine accumulation, closely tied to its terminal metabolic function. Additionally, post-inoculation with the rice blast pathogen, Spd in the knockout mutant rapidly degraded, maintaining relative stability at 12 h and 24 h, while Spm consistently decreased (Figure 7A,B). *OsPAO4* knockout mutants became more susceptible (Figure 5A–C), indicating Spm’s close association with maintaining rice blast disease resistance. Overexpression of *OsPAO4* led to a trend of initially decreasing followed by increasing levels of Spm and Spd after pathogen inoculation, suggesting their important roles in the immune response. These results indicate that both knockout and overexpression of *OsPAO4* can lead to the accumulation of Spd and Spm, which are responsive to blast fungus invasion.

Since PAs share a common precursor, S-adenosyl-L-methionine (SAM), with ethylene (ETH), alterations in polyamine levels could impact ethylene synthesis and metabolism, thereby regulating the immune response. Consequently, we conducted fluorescence quantitative analysis of genes related to the ethylene synthesis and metabolism pathways. The results indicate that the knockout of *OsPAO4* resulted in an increase in the expression level of the transcriptional repressor *OsERF3*, a decrease in ethylene production, and increased susceptibility (Figure 5A and Figure 7D). This corresponds to the downregulation of *OsACO2*, *OsACO3*, and *OsACS1* gene expression (Figure 7E). Notably, the expression of *OsACO1* was accumulated significantly in *OsPAO4* knockout lines, and the expression of *OsACO3* was inhibited in both *OsPAO4* knockout and over-expressing lines. These results indicate that the balance of *OsPAO4* expression is very important in rice resistance to *M. oryzae* through the regulation of PA accumulation and the ETH pathway.

## 3. Discussion

miRNAs, serving as either positive or negative regulators, play crucial roles in the interactions between plants and pathogens [2]. In rice, numerous miRNAs have been discovered to respond to *M. oryzae*, such as miR160a [37], miR164a [24], miR166k-miR166h [39], Osa-miR167d [25], miR169 [27], miR319b [28], miR396 [29], miR398b [23], Osa-miR439 [45], miR444b.2 [30], Osa-miR535 [32], and miR7695 [40], which have been identified as positive or negative regulators. In our previous study, we obtained 52 novel miRNAs related to rice blast fungus response from the constructed small RNA sequencing library, and degradome sequencing was used for further target gene determination for these novel miRNAs [41]. Among these novel miRNAs, Osa-miR11117 (previously named miR-T21) was selected for further characterization of its role in the regulation of rice blast resistance.

### 3.1. Impact of Osa-miR11117 on Rice Blast Resistance

Osa-miR11117 is a type of miRNA consisting of 23 bases, and its expression is found in high levels in coleoptile and leaf tissues but is comparatively lower in the stem and internode tissues compared to the roots (Figure 1A–D). This indicates that Osa-miR11117 exhibits tissue-specific expression. The responsiveness of Osa-miR11117 to *M. oryzae* invasion is different between susceptible and resistant accessions. This observation is similar to previous research findings such as Osa-miR439 [45] and Osa-miR167d [25], indicating that Osa-miR11117 is differentially responsive to *M. oryzae*. Through multiple evidence by using overexpression, target mimicry, and gene knockout of Osa-miR11117 transgenic lines, we confirmed that Osa-miR11117 negatively regulates rice blast resistance (Figure 2B–D). These findings suggest that Osa-miR11117 is highly responsive to *M. oryzae* infection and plays a significant role in the immunity of rice against the blast fungus. This responsiveness, combined with its tissue-specific expression, makes Osa-miR11117 a compelling candidate in the intricate network of rice defense mechanisms.

### 3.2. OsPAO4 Is a Direct Target of Osa-miR11117

Through bioinformatics prediction and the examination of the expression level of the candidate target genes in *N. benthamiana* leaves (Appendix A), *LOC_Os04g57550.1* (*OsPAO4*) and *LOC_Os05g51380.2* emerged as potential target genes. Indeed, the findings presented in Appendix A suggest that *LOC_Os05g51380.2*, encoding a member of the cysteine protease family proteins, could potentially be another target gene of Osa-miR11117. It is well-established that miRNAs often operate by regulating the expression of multiple target genes to fine-tune different pathways [46]. Therefore, the role of *LOC_Os05g51380.2* needs further investigation and examination. Herein, by utilizing the *GFP*-based reporter assay in *N. benthamiana* plants, we found the expression of *OsPAO4* (*OsPAO4mts-GFP*) with mutated target sites cannot be inhibited, whereas the expression of *OsPAO4* (*OsPAO4ts-GFP*) without mutations can be suppressed (Figure 4B,C), which were further confirmed with the western blot assay (Figure 4D), strongly indicating that Osa-miR11117 is a regulator of *OsPAO4*’s expression. Interestingly, in both the susceptible and resistant accessions, Osa-miR11117 and *OsPAO4* exhibited contrasting expression patterns in response to *M. oryzae* infection (Figure 1E and Figure 4A). Similar regulatory mechanisms have been elucidated in other miRNAs; for example, miR396 negatively regulates rice blast disease resistance via suppressing multiple *OsGRFs*, which in turn differentially control growth and yield [29]. In addition, Osa-miR1432 confers rice bacterial blight resistance by suppressing *OsCaML2* [47]. The ghr-miR5272a-mediated regulation of *GhMKK6* expression contributes to the immune response in cotton [48]. These reports and our results illustrate that miRNAs are widely present in the regulation of plant disease resistance and participate in plant disease resistance pathways by controlling the expression levels of target genes.

### 3.3. OsPAO4 Positively Regulates Rice Blast Resistance

Among the seven *PAO* genes in rice, *OsPAO4* is closed to *OsPAO5*, and they are both closed to *AtPAO4* (Appendix A). *OsPAO5* has been previously shown to have a detrimental effect on the lengthening of the mesocotyl by releasing less H_2_O_2_ and synthesizing more ethylene, which allows for the development of directly planted rice varieties that exhibit enhanced crop yield [43]. The four *PAO* genes (*OsPAO1*, *OsPAO3*, *OsPAO4,* and *OsPAO5*) expressed a higher level in rice leaves, and qRT-PCR analysis showed only *OsPAO4* responded to *M. oryzae* infection in rice leaves (Figure 4A and Appendix A), indicating the important roles of *OsPAO4* in blast resistance regulation. Further examination of *OsPAO4* expression in response to PA treatment showed, except Put, both Spm and Spd led to an increase in the expression of *OsPAO4* in the blast-resistant accession (Figure 4B,C), which is in accordance with the role of *OsPAO4* in Spm to Spd back-conversion processes and in PA catabolism.

Knocking out *OsPAO4* increased susceptibility to *M. oryzae* compared to the wild-type control (Figure 5A–C) and upregulated *PR* genes (*PR1a*, *PR1b*, and *PR10a*) (Figure 5D), suggesting a potential link between *R* gene *Pik-H4*-mediated blast resistance and *OsPAO4* and conversely, overexpressing *OsPAO4* enhanced resistance to *M. oryzae* (Figure 6A,B) and upregulated *PR* genes (*PR1a*, *PR1b*, and *PR10a*) (Figure 6A–E). The *ko-ospao4* lines exhibited lower levels of H_2_O_2_ compared to the wild-type (Figure 5E), while the overexpression lines displayed elevated levels of H_2_O_2_ compared to the wild-type (Figure 6F). Additionally, the activities of key ROS-scavenging enzymes, such as SOD, POD, and CAT, were significantly higher in the *ko-ospao4* lines (Figure 5F–H) while low in overexpression lines (Figure 6G–I). This suggests that the capacity to scavenge ROS in the knockout lines and overexpressing lines are contrary, which may contribute to the susceptibility and resistance against *M. oryzae* in the two lines, respectively. These findings collectively suggest that the absence of *OsPAO4* leads to an altered redox balance, with the *ko-ospao4* lines displaying a greater capacity to scavenge ROS. Importantly, after inoculation with *M. oryzae*, H_2_O_2_ levels in the *ko-ospao4* lines were notably reduced compared to the wild type (Figure 5I). This reduction in H_2_O_2_ accumulation in response to pathogen attack further underscores the compromised resistance in the absence of *OsPAO4*.

Our findings collectively underscore the important role of *OsPAO4* in rice blast resistance. The observed alterations in ROS dynamics in the absence or overexpression of *OsPAO4* suggest a potential link among PAs, ROS signaling, and plant immunity. PAs have been implicated in various stress responses in plants, and their crosstalk with ROS signaling pathways merits further investigation [13,18,22]. Additionally, the influence of *OsPAO4* on immunity suggests that it may be a key player in the early stages of plant defense responses, and these results also indicate that *R* gene-mediated ETI depends on the *OsPAO4*-mediated pathway. In conclusion, our study elucidates the multifaceted role of *OsPAO4* in modulating rice immunity against *M. oryzae*.

### 3.4. Association of OsPAO4 with Polyamine Metabolism and Ethylene Pathway

PA catabolism serves as a significant pathway for the production of H_2_O_2_ [49]. This process involves polyamine oxidases (PAOs), which generate H_2_O_2_ through the oxidative breakdown of the Put, Spd, and Spm. Among the seven *PAOs*, *OsPAO1*, *OsPAO3*, *OsPAO4*, and *OsPAO5* are responsible for catalyzing PA back-conversion reactions of PAs, including Spd, Spm, and Put [44]. This oxidation by PAOs results in the formation of the respective amino aldehydes, 1,3-diaminopropane, and H_2_O_2_ [50]. Given *OsPAO4*’s possible involvement in controlling H_2_O_2_ levels and its impact on resistance against rice blast disease, we explored the role of *OsPAO4* on PAs levels. Remarkably, both knockout and overexpression of *OsPAO4* resulted in significant accumulation of Spd and Spm (Figure 7A,B), while Put levels did not significantly change in both *OsPAO4* overexpressing and *OsPAO4* knockout plants (Figure 7C). This observation suggests that *OsPAO4* plays a pivotal role in modulating the levels of Spd and Spm, with implications for rice blast resistance. Further investigations post-inoculation with the rice blast pathogen unveiled intriguing dynamics in PA levels. In *OsPAO4* knockout mutants, Spd rapidly degraded post-inoculation but then stabilized at 12 and 24 hpi (Figure 7B). Conversely, Spm levels consistently decreased (Figure 7A). These fluctuations in polyamine levels correlated with altered disease susceptibility, reinforcing the importance of Spm in maintaining rice blast disease resistance (Figure 7A). On the other hand, overexpression of *OsPAO4* led to a transient decrease in Spd and Spm levels, followed by an increase after pathogen inoculation (Figure 7A,B). This dynamic response underscores the significance of these PAs in the rice immune response against *M. oryzae*. The connection between *OsPAO4* and polyamine metabolism is of great interest due to the potential impact on rice’s defense strategies. PAs are known to participate in various cellular processes, including stress responses. Previous research has demonstrated that Spm can function as a plant defense activator, enhancing the plant’s resistance to stress [51]. In fact, our study has shown that both Spm and Spd can enhance resistance to rice blast disease, but Spm’s enhancement is more significant (our unpublished data). Our findings suggest that *OsPAO4* is a central player in regulating polyamine levels, contributing to rice’s ability to combat *M. oryzae* infection.

Notably, PAs and ethylene share a common precursor, S-adenosyl-L-methionine (SAM), in their biosynthesis pathways [52]. Alterations in polyamine levels could thus influence ethylene synthesis and metabolism, subsequently modulating the plant’s immune response. Our results revealed that knockout of *OsPAO4* resulted in an upregulation of the transcriptional repressor *OsERF3*, accompanied by a decrease in ethylene production [53]. This shift in ethylene dynamics was associated with increased susceptibility to *M. oryzae*. Furthermore, this trend corresponded with the downregulation of key ethylene synthesis-related genes, including *OsACO2*, *OsACO3*, and *OsACS1* (Figure 7D).

These findings suggest that *OsPAO4* plays a crucial role in regulating rice’s resistance to *M. oryzae* by controlling PA accumulation and modulating the ETH pathway. The crosstalk between PAs and ethylene, both of which are pivotal in plant stress responses, provides a compelling avenue for further research into the intricate network of signaling pathways involved in rice immunity.

## 4. Methods

### 4.1. Plant Materials and Growth Conditions

Rice (*Oryza sativa* L.) plants used in this study were the blast-susceptible japonica accession Lijiangxintuanheigu (LTH) and the blast-resistant accession Pik-H4 NIL (including disease resistance gene Pik-H4) [54]. All these rice plants were grown in a growth chamber at a temperature of 28 °C and 70% relative humidity with 14 h light, followed by 10 h dark, while the N. benthamiana plants, used for Agrobacterium-infiltration experiments, were maintained in a growth chamber at a temperature of 22 °C and 14 h light, followed by 10 h dark.

### 4.2. Plasmid Construction and Genetic Transformation

To create the Osa-miR11117 overexpression (OX11117), Osa-miR11117 target mimicry (MIM11117), and *OsPAO4* overexpression (OX11117) plasmid constructs, pOX plasmid was utilized [55]. Osa-miR11117 precursor sequence, target mimicry sequence, and coding sequence of *OsPAO4* were cloned into *BamH*I–*Hind*III; sites of the binary vector pOX plasmid, respectively. For the CRISPR/Cas9 plasmids of KO11117 and *ko-ospao4*, target primers (Appendix A) of Osa-miR11117 and *OsPAO4* were designed respectively according to the CRISPR/Cas9 online target design website (http://crispr.hzau.edu.cn/CRISPR2/ accessed on 20 August 2019). Then, using the pGRT vector as a template, PCR amplification was performed to obtain the gRNA-target sequence-tRNA module, which was then cloned into the Cas9 protein expression vector pRGEB32, respectively. After colony PCR and subsequent sequencing verification of the above constructs, the positive transformants were transformed into rice blast-resistant accession Pik-H4 NIL through *Agrobacterium*-mediated methods, respectively.

For the reporter plasmids, we introduced artificial sequences (*OsPAO4ts* and *OsPAO4mts*) of the Osa-miR11117 target gene site at the beginning of the eGFP codon sequences. These constructs and the Osa-miR11117 precursor sequence were then inserted into the *Kpn*I site of the binary vector pEXT06G1, respectively. These constructed plasmids were subsequently employed in *Agrobacterium*-mediated transient expression assays in *N. benthamiana* plants.

### 4.3. Pathogen Growth, Infection, and PAs Treatment

In this study, we utilized two strains of *M. oryzae*, namely Guy11 and GDYJ7. To cultivate these *M. oryzae* strains, we used a complete medium (CM) as described by Chandran et al. [29]. These strains were cultured at 28 °C under a 12 h light/12 h dark cycle to induce sporulation. After two weeks of cultivation, spores were collected, and their concentration was adjusted to 5 × 10^5^ spores mL^−1^ for the subsequent assays. For spray inoculation, four-leaf-stage seedlings were inoculated with the spores. The disease phenotypes on the second leaf were recorded at 6 dpi. In the punch inoculation method [56], 5 µL of the spore suspension was applied at two spots on each leaf, and the leaves were kept in a culture dish containing 0.1% 6-Benzylaminopurine (6-BA). Lesion length was measured after 6 dpi. We calculated the relative fungal mass by comparing the DNA concentration of *M. oryzae Pot2* against the rice genomic ubiquitin DNA level using qPCR, following the protocols by Li et al. [27].

For PAs treatment, different concentrations of Spd (10 mM), Spm (10 mM), and Put (10 mM) were sprayed on four-leaf-stage seedlings, and samples were collected at indicated time points, respectively. Similar results were obtained from three independent experiments.

### 4.4. GUS Staining Assay

An approximately 2 kb promoter sequence of Osa-miR11117 precursor was cloned and inserted into the pCAMBIA1305 vector. The resulting P_Osa-miR11117_-*GUS* plasmid was then introduced into *Agrobacterium tumefaciens* strain EHA105 for infection of the wild-type Pik-H4 NIL callus to obtain the P_Osa-miR11117_: *GUS* transgenic rice plants. Subsequently, the transgenic plant tissues were subjected to a series of washes using a 100 mM Na_3_PO_4_ buffer (pH 7.0) three times. Following the washes, the tissues were incubated with a GUS-staining solution. This solution contained 100 mM Na_3_PO_4_ (pH 7.0), 10 mM EDTA, 2 mM 5-bromo-4-chloro-3-indolyl-β-GlcA, 5 mM K_4_Fe(CN)_6_, 5 mM K_3_Fe(CN)_6_, and 0.2% Triton X-100. The incubation was carried out at a temperature of 37 °C, ranging from 20 min to 24 h, as per the method described by Jefferson et al. [57]. Similar results were obtained from three independent experiments.

### 4.5. RNA Isolation, Quantitative RT-PCR, and Northern Blot

To isolate total RNA from collected samples, we used TRIzol reagent (Thermo Fisher Scientific, Waltham, MA, USA). Reverse transcription to generate cDNA was performed using the SuperScript first-strand synthesis kit (Invitrogen). The resulting cDNA was used for the qPCR reactions, which were performed using SYBR Green Real-Time PCR Master Mix (Toyobo, Osaka, Japan), following the manufacturer’s instructions. For miRNA expression analysis, we employed the stem-loop pulse qRT-PCR method [58], utilizing designed stem-loop primers for reverse transcription and forward primers containing the 5′ end of mature miRNA sequences (both forward and reverse primers) (Appendix A) based on the design by Varkonyi-Gasic et al. [58]. For internal reference, U6 snRNA-specific primers were used. To normalize the expression of target genes, the rice ubiquitin (UBQ) gene served as the internal reference. Relative expression of the target genes was calculated using the 2^–ΔΔCT^ method. Each experiment was carried out with three biological repeats [54].

An amount of 3 μg of total RNA was separated on a 12% denaturing polyacrylamide gel and then transferred to Hybond-N+ membranes (GE, Amersham, Boston, MA, USA) at a current of 400 mA for 1 h. To crosslink the RNA, UV light was applied using the HL-2000 HybriLinker (UVP) for 2 min. Following this, the membrane was placed in Dig Easy Hyb solution (Roche, Basel, Switzerland) at 42 °C for 1 h, and subsequently, it was hybridized with the 5′-Dig labeled miRNA probe in the same hybridization buffer, with incubation at 42 °C overnight. The membrane was then washed using the Dig Wash and Block Buffer Set (Roche) as per the manufacturer’s instructions.

### 4.6. Agrobacterium-Mediated Transient Expression Analysis in N. benthamiana

*Agrobacterium* strain GV3101 containing the individual constructs was cultured at 28 °C overnight in LB media supplemented with kanamycin (50 mg/mL), rifampin (50 mg/mL), and gentamicin (50 mg/mL) in a shaker. The bacteria were then collected by centrifugation, and the pellet was resuspended in an MMA buffer (consisting of 10 mM MgCl_2_, 10 mM MES, and 200 mM acetosyringone). Subsequently, the Agrobacteria containing the constructs were introduced into the leaves of *N. benthamiana* by infiltration using a 1 mL syringe without a needle for a transient expression assay. The fluorescence signal was observed at 48 hpi using a confocal microscope. Simultaneously, samples were collected, and a western blot assay was conducted. Briefly, 100 mg of fresh *N. benthamiana* leaves were used to extract total protein with a protein extraction buffer (comprising 0.25 M Tris-HCL, pH 6.8; 4% SDS; 0.1% bromophenol blue; and 40% glycerol). The total proteins were separated using 12% SDS-PAGE and transferred onto a PVDF membrane (Millipore, Darmstadt, Germany) using Trans-Blot Turbo (BIO-RAD, Hercules, CA, USA). The polyclonal anti-GFP antibody (1:2000) was used to interact with the target protein, and the Clarity™ Western ECL Substrate system (BIO-RAD) was employed to detect GFP accumulation. Similar results were obtained from three independent experiments.

### 4.7. H_2_O_2_ Measurement and Enzymatic Activity Assay

To assess H_2_O_2_ levels and the activities of the enzymes SOD, POD, and CAT, 100 mg of fresh leaf samples by homogenizing them with 900 µL of PBS (pH 7.4, 0.1 M), according to the ratio of weight (g):volume (mL) = 1:9, using a glass homogenizer. After centrifugation at 3500× *g* for 12 min, the supernatant was used for analysis. Commercially available test kits from Nanjing Jiancheng Bioengineering Institute, Nanjing, China, were employed for the assays.

H_2_O_2_ content was determined using the H_2_O_2_ Assay kit (A064-1-1), following the manufacturer’s instructions. H_2_O_2_ forms a complex with molybdate, which was quantified by measuring absorbance at 405 nm and expressed as mmol gprot^−1^. SOD activity was determined using the SOD Assay kit (A001-1), following the manufacturer’s instructions. Following thorough mixing, the samples were incubated at 37 °C for 20 min, and subsequently, 200 μL of the mixture was extracted and placed in a 96-well enzyme standard plate. The absorbance was then measured at 450 nm. SOD activity was defined as one unit, corresponding to the amount of enzyme required to achieve a 50% inhibition rate of SOD, and expressed as U mgprot^−1^. POD activity was measured using the POD Assay Kit (A084-3-1) with reagents added as per the manufacturer’s instructions. After thorough mixing, the samples were centrifuged at 3500 rpm for 10 min, and 200 μL of the resulting supernatant was transferred to a 96-well enzyme standard plate. Subsequently, the absorbance was recorded at 420 nm and expressed as U mgprot^−1^. CAT activity was assessed using the CAT Assay Kit (A007-1), and reagents were added according to the manufacturer’s protocol. After mixing, 200 μL was added to a 96-well enzyme standard plate. The absorbance was measured at 405 nm and expressed as U mgprot^−1^.

The soluble protein content was also determined using the protein quantitative test box (A045-4-2). Three experimental replicates and three biological replicates were performed for each sample. According to the instructions, the optical density (OD) value of each reaction mixture at 562 nm wavelength was determined by a microplate spectrophotometer, and the soluble protein content was calculated by combining the formula. In brief, H_2_O_2_ content (mmol gprot^−1^) = (measured OD value − blank OD value)/(standard OD value − blank OD value) × standard substance concentration /Protein concentration of the sample to be measured. POD activity (U gprot^−1^) = (determination OD − control OD)/(12 × cupola light diameter) × (1000 × total volume of reaction liquid (mL))/sample size (mL)/reaction time/protein concentration of the sample to be measured. SOD activity (U gprot^−1^) = (control OD value − measured OD value)/control OD value/ 50% × total volume of reaction liquid (mL)/sample amount (mL)/protein concentration of the sample to be measured. CAT activity (U gprot^−1^) = (determination OD − control OD) × 235.65 × 1/60 × sample size (mL)/protein concentration of the sample to be measured (265.65 is the slope derivative).

### 4.8. Extraction and Analysis of PAs

PAs, including Spm, Spd, and Put, were quantified using a modified method based on Zapata et al. [59]. Leaf samples weighing between 100 and 200 mg were ground into powder using liquid nitrogen and then immersed in 1 mL of 5% trichloroacetic acid (TCA) at 4 °C for 1 h. Afterward, the samples underwent centrifugation at 12,000 rpm at 4 °C for 30 min, and 500 μL of the resulting supernatant was transferred to a new 15-mL tube.

To this supernatant, 7 μL of benzoyl chloride and 1 mL of 2 mol L^−1^ NaOH were added, followed by vortexing for 20 s. The mixture was then incubated at 37 °C for 20 min. Subsequently, 2 mL of saturated NaCl and 1.5 mL of ethyl ether were introduced, and the mixture was centrifuged at 1500× *g* for 10 min at 4 °C. In a new tube, 1 mL of the ether phase was carefully collected and subjected to vacuum drying. The resulting residue was dissolved in 150 μL of high-performance liquid chromatography (HPLC) grade methanol.

For HPLC analysis, an ODS-BP column (5 μm, 4.6 mm × 200 mm, Elite, Shenzhen, China) was used in a system consisting of a ddH_2_O pump-A and a gradient (acetonitrile) pump-B, along with a UV detector. A total of 20 μL of samples were injected, and PAs were eluted using a mobile phase composed of acetonitrile and ddH_2_O (in a 50:50 ratio, *v*/*v*) at a flow rate of 1 mL per minute. The benzoyl-PAs were detected via absorbance at 254 nm. The results are presented as μg per gram of fresh weight (μg g^−1^ FW) and represent the mean ± SE of extractions from three distinct samples for each treatment. Each experiment was carried out with three biological repeats.

## 5. Conclusions

Osa-miR11117 negatively regulates blast resistance by targeting *OsPAO4* to suppress its expression and maintain Spd and Spm at a relatively stable level to cope with changes in both internal and external environments. Our findings suggest that *OsPAO4* may regulate rice blast resistance by influencing PA metabolism and the ethylene signaling pathway. The findings provide valuable insights into the complex interplay between polyamine metabolism, ROS signaling, and plant defense mechanisms. Further research is warranted to unravel the precise molecular mechanisms underlying *OsPAO4*-mediated resistance and its potential applications in enhancing rice blast resistance in agricultural settings.

## Figures and Tables

**Figure 1 ijms-24-16052-f001:**
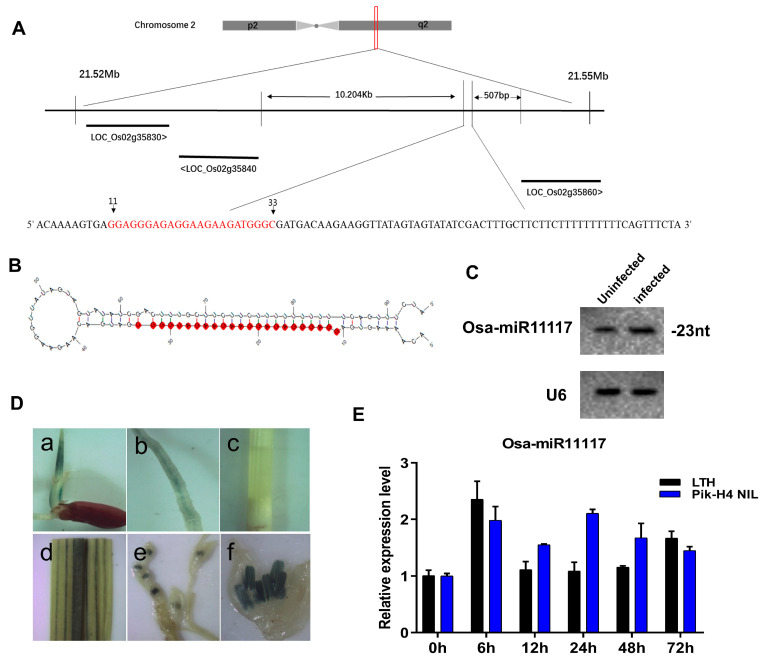
Validation of the novel miRNA Osa-miR11117, the promotor activity of Osa-miR11117, and Osa-miR11117′s response to *M. oryzae.* (**A**) The precursor locus of Osa-miR11117 in chromosome 2. The red-highlighted sequence indicates the position of Osa-miR11117 within the precursor sequence. (**B**) Structure of the miRNA precursor. The red-highlighted portion indicates the position of the mature Osa-miR11117 sequence within the secondary structure of the precursor sequence. (**C**) Northern blotting analysis of Osa-miR11117 in leaves of resistant accession Pik-H4 NIL infected with *M. oryzae*. (**D**) *GUS* assay for the promotor activity of Osa-miR11117. (**a**) plumule; (**b**) root; (**c**) stem; (**d**) leaf; (**e**) spikelet; (**f**) anther. (**E**) Accumulation of Osa-miR11117 was performed using stem-loop qRT-PCR in susceptible and resistant accessions upon *M. oryzae* (or mock) infection. Values are means of three replications. Error bars indicate SD.

**Figure 2 ijms-24-16052-f002:**
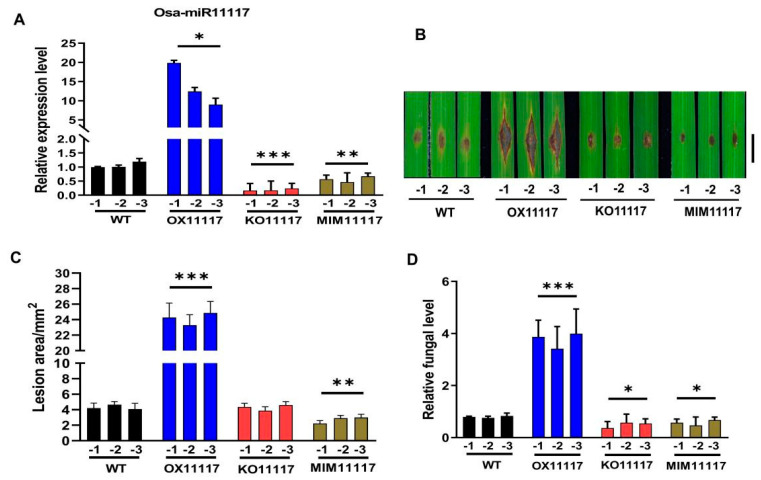
Resistance of rice plants with overexpression, knockout, and target mimic of Osa-miR11117 to *M. oryzae*. (**A**) The expression of Osa-miR11117 in OX11117, KO11117, and MIM11117 transgenic plants, WT represents the wild-type rice accession Pik-H4 NIL. OX11117, KO11117, and MIM11117 represent the overexpression, knockout, and target mimicry overexpression transgenic lines of Osa-miR11117, respectively. (**B**) Disease phenotypes of OX11117, KO11117, and MIM11117 transgenic plants, respectively. Representative images of *M. oryzae*-infected leaves at 6 dpi. (**C**) Quantification of blast lesions was carried out by image analysis. (**D**) Relative quantification of fungal biomass was determined by qPCR using specific primers for the *M. oryzae Pot2* gene (values are fungal DNA levels normalized against the rice *Ubiquitin 1* gene. Error bars indicate SD. Asterisks represent significance difference determined by Student’s *t* test (* *p* < 0.05; ** *p* < 0.01; *** *p* < 0.001) compared with the WT control. Similar results were obtained from three independent experiments.

**Figure 3 ijms-24-16052-f003:**
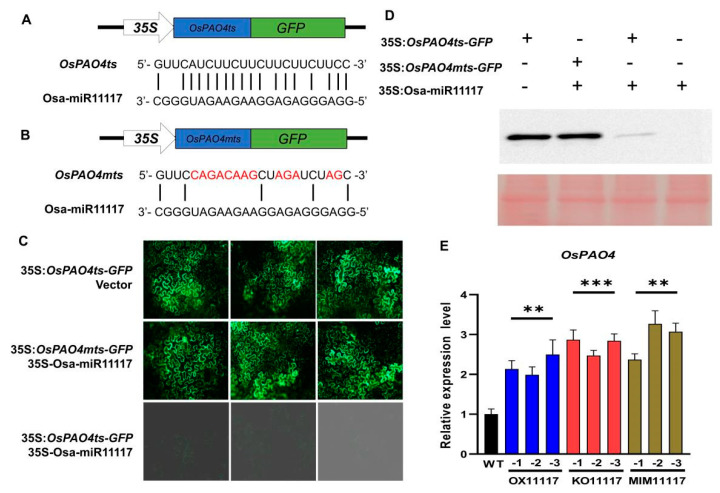
Verification of *OsPAO4* as a target gene regulated by Osa-miR11117. (**A**) The construct of fused-green fluorescent protein (GFP) with the target site of *OsPAO4* (*OsPAO4ts-GFP*) and alignment of Osa-miR11117 with *OsPAO4ts* target site sequences. (**B**) The construct of fused GFP with the mutated target site of *OsPAO4* (*OsPAO4mts-GFP*) and alignment of Osa-miR11117 with *OsPAO4mts* target site sequences. The red letters indicate the mutated bases. (**C**) The protein concentrations of OsPAO4ts-GFP and OsPAO4mts-GFP. The indicated *OsPAO4ts-GFP* and *OsPAO4mts-GFP* reporter constructs were transiently expressed alone or coexpressed with Osa-miR11117 in *N. benthamiana* leaves using *Agrobacterium*-mediated infiltration. (**D**) Western blotting analysis shows the protein concentrations of OsPAO4ts-GFP and OsPAO4mts-GFP. Ponceau S staining was used for total protein normalization. (**E**) The expression of *OsPAO4* in the indicated lines of OX11117, KO11117, and MIM11117 transgenic plants, respectively. Values are means of three replications. Error bars indicate SD. Asterisks represent significance difference determined by Student’s *t* test (** *p* < 0.01; *** *p* < 0.001) compared with the WT control. Similar results were obtained from three independent experiments.

**Figure 4 ijms-24-16052-f004:**
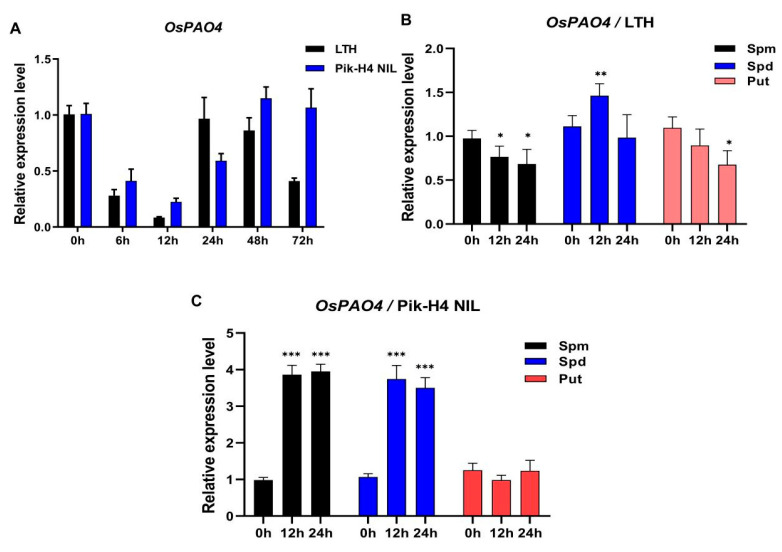
*OsPAO4* is one *M. oryzae* and PA responsive *PAO* gene. (**A**) The expression of *OsPAO4* in susceptible and resistant accessions upon *M. oryzae* GDYJ7 infection. Values are means of three replications. Error bars indicate SD. (**B**,**C**) The expression of *OsPAO4* in susceptible and resistant accessions after PA (Spm, Spd, and Put) treatment at 0 h, 12 h, and 24 h. Error bars indicate SD. Asterisks represent significance differences determined by Student’s *t* test (* *p* < 0.05; ** *p* < 0.01; *** *p* < 0.001) compared with the 0 h samples. Similar results were obtained from three independent experiments.

**Figure 5 ijms-24-16052-f005:**
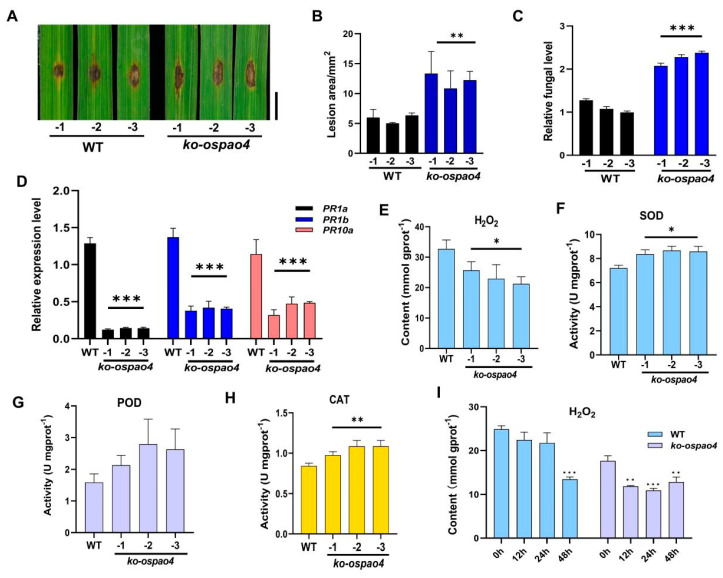
Knockout of *OsPAO4* enhances the susceptibility to infection by *M. oryzae* by regulating the *PR* genes expression and the accumulation of H_2_O_2_. (**A**) Blast disease phenotypes on leaves at 6 days post-inoculation of *M. oryzae* strain GDYJ7 by punch inoculation. Bar = 5 mm. (**B**) The lesion area of the rice leaves (WT and *ko-ospao4*) corresponding to (**A**). (**C**) Relative fungal biomass of GDYJ7 on WT and *ko-ospao4* leaves. The relative fungal biomass was measured by using the ratio of the DNA level of the *M. oryzae Pot2* gene against the rice *Ubiquitin 1* level. (**D**) The expression of *PR* genes (*PR1a*, *PR1b*, and *PR10a*) in *OsPAO4* knockout mutants. (**E**–**H**) The accumulation of H_2_O_2_ and the enzymatic activity of SOD, POD, and CAT in *OsPAO4* knockout mutants. (**I**) The accumulation of H_2_O_2_ of four-leaf-stage in leaves of *ko-ospao4* upon *M. oryzae* GDYJ7 infection. Asterisks represent significance differences determined by Student’s *t* test (* *p* < 0.05; ** *p* < 0.01; *** *p* < 0.001) compared with the WT control or 0 h samples. Similar results were obtained from three independent experiments.

**Figure 6 ijms-24-16052-f006:**
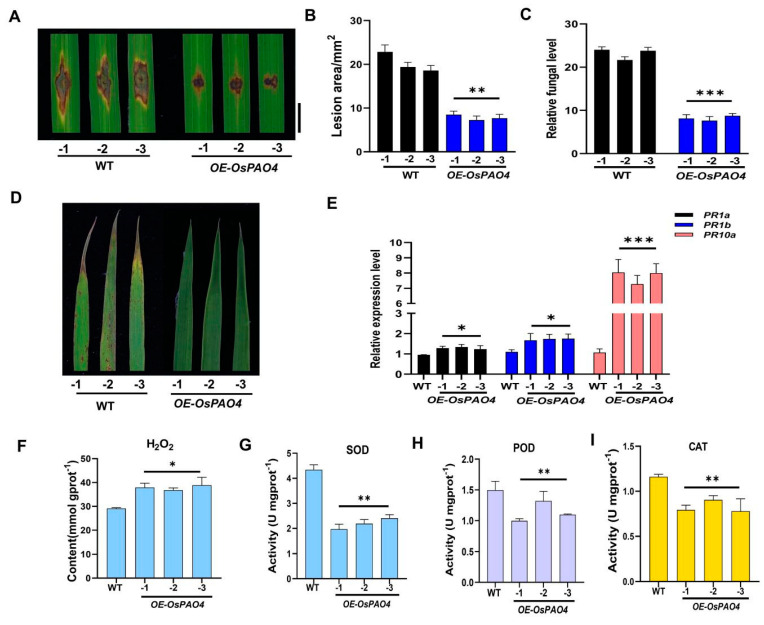
Overexpression of *OsPAO4* enhances the resistance against *M. oryzae* by regulating the *PR* genes expression and the accumulation of H_2_O_2_. (**A**) Blast disease phenotypes on leaves at 6 dpi of *M. oryzae* strain Guy11 by punch inoculation. Bar = 5 mm. (**B**) The lesion area of the rice leaves (WT and *OE-OsPAO4*) corresponding to (**A**). (**C**) Relative fungal biomass of Guy11 on WT and *OE-OsPAO4* leaves. (**D**) Blast disease phenotypes on leaves at 7 dpi of *M. oryzae* strain Guy11 by spray inoculation. (**E**) The expression of *PR* genes (*PR1a*, *PR1b*, and *PR10a*) in *OsPAO4* overexpressing lines. (**F**–**I**) The accumulation of H_2_O_2_ and the enzymatic activity of SOD, POD, and CAT in *OsPAO4* overexpressing lines. Asterisks represent significance difference determined by Student’s *t* test (* *p* < 0.05; ** *p* < 0.01; *** *p* < 0.001) compared with the WT control. Similar results were obtained from three independent experiments.

**Figure 7 ijms-24-16052-f007:**
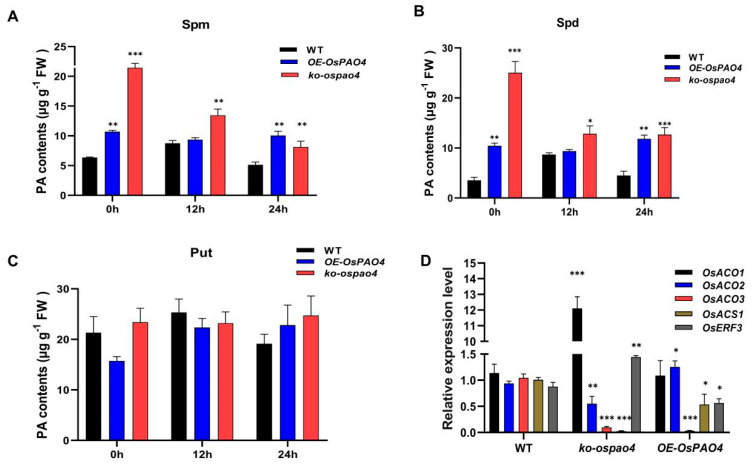
*OsPAO4* regulates rice resistance to *M. oryzae* by controlling both the accumulation of PA and the ethylene signaling pathway. (**A**–**C**) The levels of PA (Spd, Spm, and Put) in *OsPAO4* overexpressing lines and *OsPAO4* knockout mutants. (**D**) The expression of ethylene signaling genes (*OsACO1*, *OsACO2*, *OsACO3*, *OsACS1*, and *OsERF3*) in *OsPAO4* knockout and overexpressing lines. Asterisks represent significance differences determined by Student’s *t* test (* *p* < 0.05; ** *p* < 0.01; *** *p* < 0.001) compared with the WT control or 0 h samples. Similar results were obtained from three independent experiments.

## Data Availability

The data presented in this study are available within the article text and figures.

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
