# Peer review of "Osa-miR11117 Targets OsPAO4 to Regulate Rice Immunity against the Blast Fungus Magnaporthe oryzae"

_ijms, 2023, doi:10.3390/ijms242216052_

Round 1

Reviewer 1 Report

Comments and Suggestions for Authors

Gao et al submitted a manuscript titled "Osa-miR11117 targets OsPAO4 to regulates rice immunity against the blast fungus Magnaporthe oryzae", for publication in IJMS.  

The conclusion of this work, "OsPAO4 may regulate rice blast resistance by influencing PA metabolism and the ethylene signaling pathway" is interesting. The work is done well and written well. The ms can be accepted for publication provided these minor comments are addressed.

1. Why past tense in "2.1. Osa-miR11117 was responsive to blast fungus"?

2. Cite "UNAFold analysis" in page 3 of 17.

3. In almost all figures, the statstical tests are missing. The figures show "stars", but fails to mention everywhere the tests used.

Comments on the Quality of English Language

NA

Author Response

The conclusion of this work, "OsPAO4 may regulate rice blast resistance by influencing PA metabolism and the ethylene signaling pathway" is interesting. The work is done well and written well. The ms can be accepted for publication provided these minor comments are addressed.

  1. Why past tense in "2.1. Osa-miR11117 was responsive to blast fungus"?

Thank you for pointing this out. We have changed past tense to simple present tense in "2.1. Osa-miR11117 is responsive to blast fungus".

  1. Cite "UNAFold analysis" in page 3 of 17.

Thank you for this suggestion. We have added corresponding reference for the "UNAFold analysis" in page 3 of 17.

  1. In almost all figures, the statstical tests are missing. The figures show "stars", but fails to mention everywhere the tests used.

Thank you for pointing out this issue. We have supplemented the method of statistical analysis and the meaning of asterisks in the corresponding figure legends. Asterisks represent significance difference determined by Student’s t test (*P < 0.05; **P < 0.01; ***P < 0.001) compared with the WT control or the 0 h samples. Similar results were obtained from three independent experiments.

Reviewer 2 Report

Comments and Suggestions for Authors

The work presented by Gao et al., entitled “Osa-miR11117 targets OsPAO4 to regulates rice immunity against the blast fungus Magnaporthe oryzae” has certainly very importance for researchers working on plant pest management. The manuscript is well written and described in terms of results and discussion. This article can be accepted in this form after few minor corrections.

1-    In methodology section, authors can describe more detail about procedure/protocols e.g., H2Omeasurement that would be easier to understand for audience.

2-    Authors are encouraged to write more about PAs and its role in plant immunity especially in introduction and discussion part citing with recent articles.

3-    There are a very few English errors in the MS that need to be thoroughly revised by authors.

Comments on the Quality of English Language

There are a very few English errors in the MS that need to be thoroughly revised by authors.

Author Response

The work presented by Gao et al., entitled “Osa-miR11117 targets OsPAO4 to regulates rice immunity against the blast fungus Magnaporthe oryzae” has certainly very importance for researchers working on plant pest management. The manuscript is well written and described in terms of results and discussion. This article can be accepted in this form after few minor corrections.

  1. In methodology section, authors can describe more detail about procedure/protocols e.g., H2O2 measurement that would be easier to understand for audience.

Thanks for your kind reminding. We have added more detail for H2O2 measurement in methodology section.

  1. Authors are encouraged to write more about PAs and its role in plant immunity especially in introduction and discussion part citing with recent articles.

Thank you for this suggestion. We have added more progress about PAs and its role in plant immunity in introduction and discussion part.

  1. There are a very few English errors in the MS that need to be thoroughly revised by authors.

Thank you for pointing this out. We have thoroughly revised the MS for the English errors.

  1. Comments on the Quality of English Language

There are a very few English errors in the MS that need to be thoroughly revised by authors.

Thank you for pointing this out. We have thoroughly revised the MS for the English errors.

Reviewer 3 Report

Comments and Suggestions for Authors

the paper is very interesting with good amount of results but some points need to be improved: 

Abstract need to be re-write with: sentence of general introduction, main of the work with material and methods, main results, and finish with sentence of conclusion.

Material and methods

This part is well described but part statistic need to be written in the manuscript and how much replication for how much plants and how much you repeat the experimental, all of that need to be clarify

The results

In all the histograms of the figures there is an Asterisk, what does it mean?

And one figure there are many panels and in the text isn’t well described also in the caption, please improve is very important

In the part discussion the author mainly repeat their results so please improve the part discussion

Author Response

the paper is very interesting with good amount of results but some points need to be improved:

1.Abstract need to be re-write with: sentence of general introduction, main of the work with material and methods, main results, and finish with sentence of conclusion.

Thank you for this point. We have revised the abstract according to your suggestion.

2.Material and methods:This part is well described but part statistic need to be written in the manuscript and how much replication for how much plants and how much you repeat the experimental, all of that need to be clarify.

Thank you for pointing this out. We have added the information for the replicates and plants used in the Material and methods part of this MS.

3.In all the histograms of the figures there is an Asterisk, what does it mean?

Thank you for pointing this out. Asterisks represent significance difference determined by Student’s t test ( *P < 0.05; **P < 0.01; ***P < 0.001) compared with the WT control or 0 h time-point sample. Similar results were obtained from three independent experiments. 

4.And one figure there are many panels and in the text isn’t well described also in the caption, please improve is very important

Thank you for this suggestion. We have revised the corresponding caption to make it more accurately.

  1. In the part discussion the author mainly repeat their results so please improve the part discussion.

Thank you for pointing this out. We have revised the discussion part.